# GridHTM: Grid-Based Hierarchical Temporal Memory for Anomaly Detection in Videos

**DOI:** 10.3390/s23042087

**Published:** 2023-02-13

**Authors:** Vladimir Monakhov, Vajira Thambawita, Pål Halvorsen, Michael A. Riegler

**Affiliations:** 1SimulaMet, 0167 Oslo, Norway; 2Department of Informatics, University of Oslo, 0316 Oslo, Norway; 3Department of Computer Science, Oslo Metropolitan University, 0130 Oslo, Norway; 4Department of Computer Science, UiT The Arctic University of Norway, 9037 Tromsø, Norway

**Keywords:** HTM, deep learning, surveillance, anomaly detection

## Abstract

The interest in video anomaly detection systems that can detect different types of anomalies, such as violent behaviours in surveillance videos, has gained traction in recent years. The current approaches employ deep learning to perform anomaly detection in videos, but this approach has multiple problems. For example, deep learning in general has issues with noise, concept drift, explainability, and training data volumes. Additionally, anomaly detection in itself is a complex task and faces challenges such as unknownness, heterogeneity, and class imbalance. Anomaly detection using deep learning is therefore mainly constrained to generative models such as generative adversarial networks and autoencoders due to their unsupervised nature; however, even they suffer from general deep learning issues and are hard to properly train. In this paper, we explore the capabilities of the Hierarchical Temporal Memory (HTM) algorithm to perform anomaly detection in videos, as it has favorable properties such as noise tolerance and online learning which combats concept drift. We introduce a novel version of HTM, named GridHTM, which is a grid-based HTM architecture specifically for anomaly detection in complex videos such as surveillance footage. We have tested GridHTM using the VIRAT video surveillance dataset, and the subsequent evaluation results and online learning capabilities prove the great potential of using our system for real-time unsupervised anomaly detection in complex videos.

## 1. Introduction and Motivation

As the global demand for security and automation increases, many seek to use video anomaly detection systems. In the US alone, the surveillance market is expected to reach USD 23.60 billion by 2027 [1]. Leveraging modern computer vision, modern anomaly detection systems play an important role in increasing monitoring efficiency and reducing the need for expensive live monitoring. Their use cases can vary from detecting faulty products on an assembly line to detecting car accidents on a highway.

The most important component in video anomaly detection systems is the algorithm behind it. These can range from simple on-board algorithms [2,3] to advanced deep learning models [4,5,6], with the latter experiencing an increase in popularity over the past few years. Yet, despite major progress within the field of deep learning, there are still many tasks where humans outperform models, especially in anomaly detection where the anomalies are often undefined. In addition, deep learning approaches have challenges when dealing with online learning, noise and concept drift [7,8,9,10].

The cause for the discrepancy often lies in the difference between how humans and machine learning algorithms represent data and learn. Most machine learning algorithms use a dense representation of the data and apply back-propagation in order to learn. It is believed that human learning occurs in the neocortex, which utilizes a sparse representation and employs Hebbian-style learning, as supported by evidence [11]. For the latter, there is a growing field of machine learning dedicated to replicating the inner mechanics of the neocortex, namely Hierarchical Temporal Memory (HTM) theory [12]. HTM bears several advantages over standard machine learning, such as noise-tolerance and the ability to adapt to data changing over time (e.g., type anomalies change, behaviour changes, etc.).

The main motivation behind this work is to explore the use of HTM for visual content. In light of the advantages offered by HTM and the rise of video anomaly detection, a natural question would be whether HTM could be applied to video anomaly detection for surveillance videos.

In this paper, we therefore propose and evaluate Grid-based Hierarchical Temporal Memory (GridHTM) which is a novel expansion of the base HTM algorithm that allows for unsupervised anomaly detection in videos. The main contributions of our work are the following:We provide the theoretical and practical foundation for making HTM usable in a visual analysis context.We present GridHTM, a method that allows users to perform unsupervised anomaly detection in real-time for visual content.We show the potential of GridHTM using the VIRAT publicly available real-world dataset.We analyse and discuss possible shortcomings and future directions.We provide our implementation as open source code.

Our initial experimental results show the potential of the presented GridHTM for unsupervised visual anomaly detection in real-time. Since GridHTM can perform in real-time on little data and adjust itself to possible changes over time, it holds potential as a feasible alternative to methods that require a large amount of labelled data and are limited to offline learning schemes.

## 2. Related Work

Anomaly detection is often defined as detecting data points that deviate from the general distribution [5]. A subset of anomaly detection is smart surveillance [13], which is the use of video analysis specifically in surveillance. Unlike most other problems in deep learning, anomaly detection deals with unpredictable and rare events which makes it hard to apply traditional deep learning for anomaly detection. Popular approaches therefore often employ generative models, that calculate an anomaly measure using generated or reconstructed data [5,14,15,16]. This approach is based on the assumption that the model will only be able to generate data similar to what it has been trained on, and therefore fail when an anomalous event occurs.

However, deep learning has issues that make it difficult to apply for complex anomaly detection in surveillance scenarios that change over time and require online learning. One issue for deep-learning models in general is that they are susceptible to noise in the dataset [9,10], which leads to decreased model accuracy and poor prediction results. Likewise, concept drift is an ongoing challenge in deep learning [7,8] as deep learning models typically do not perform online learning and must be frequently adjusted to maintain performance on evolving data. Although anomalies typically do not imply a shift in the underlying data, an anomaly detection model that neglects concept drift will eventually begin detecting false anomalies.

As mentioned before, deep learning does not work well for online learning which is required in dynamic and swiftly changing environments such as video or network traffic surveillance [17]. In addition, deep learning models require a large amount of data before they can be considered effective, and performance increases logarithmically based on the volume of training data [18]. Deep learning models also suffer from issues with out-of-distribution generalization [19], where a model might perform great on the dataset it is tested on, but performs poorly when deployed in real life. This could be caused by selection bias in the dataset or when there are differences in the causal structure between the training domain and the deployment domain [20]. Another challenge with deep learning models is that they generally suffer from a lack of explainability [21]. While it is known *how* the models make their decisions, their huge parametric spaces make it unfeasible to determine *why* they make those predictions. In addition, the possible application of machine learning in critical sectors, such as medicine, makes approaches that provide some degree of explanation necessary.

In this work, we want to open up a possible new direction of online, real-time learning for visual content based on the HTM theory. The HTM theory [12] introduces a machine learning algorithm that works on the same principles as the human brain and therefore solves some of the issues that deep learning has. Here, we give a high-level overview, and we refer to original works [12,22] for more information. HTM is considered noise resistant and can perform online learning, meaning that it learns as it observes more data. HTM replicates the structure of the neocortex which is made up of cortical regions, and which in turn are made up of mini-columns and then neurons.

The data in an HTM model is represented using a Sparse Distributed Representation (SDR), which is a sparse bit array. An encoder converts real world values into SDRs, and there are currently encoders for numbers, geospatial locations, categories, and dates. One of the difficulties with HTM is making it work on visual data, where creating a good encoder for visual data is still being researched [23,24,25].

The learning mechanism consists of two parts, the Spatial Pooler (SP) and the Temporal Memory (TM). The SP learns to extract semantically important information into output SDRs. The TM learns sequences of patterns of SDRs and forms a prediction in the form of a predictive SDR. Both the SP and TM learn by growing and strengthening/weakening synapses, similar to Hebbian learning [26]. Finally, the predictive SDR can be used in tandem with a simple classifier to make a classification, or be compared against the actual outcome in order to calculate an anomaly score. The method for determining the anomaly score involves comparing the current predictive SDR and the output of the SP in the following timestep, and computing the ratio of the number of overlapping active bits to the total number of bits. Figure 1 shows the HTM pipeline at a glance.

Ahmad et al. [22] showed that HTM is highly capable of performing anomaly detection on low-dimensional data and is able to outperform other anomaly detection methods. On the other hand, related works [25] show that HTM struggles with higher dimensional data, such as video. Therefore, a natural conclusion is that HTM should be applied differently, and that a new type of architecture using HTM should be explored for the purpose of video anomaly detection and surveillance.

Overall, from the related work, we can see that HTM has potential to be extended for video anomalies detection in real-time. Nevertheless, there are still many basic open questions that warrant exploration. Thus, a comparison of HTM with other algorithms at this stage is not of interest, but rather an understanding of the internal processes and limitations is needed.

## 3. GridHTM

This paper proposes and explores a new type of architecture, named Grid-based Hierarchical Temporal Memory (GridHTM), for anomaly detection in videos using HTM, and proposes the use of segmentation techniques to simplify the data into an SDR-friendly format. These segmentation techniques could be anything from simple binary thresholding to deep learning instance segmentation. Even keypoint detectors such as Oriented FAST and Rotated BRIEF (ORB) [27] could in theory be applied. When explaining Grid-based Hierarchical Temporal Memory (GridHTM), the examples will be taken from deep learning instance segmentation of cars on a video from the VIRAT [28] dataset. An example segmentation is shown in Figure 2.

The idea is that the SP will learn to find an optimal general representation of cars. How general this representation is can be configured using the various SP parameters, but ideally they should be set so that different cars will be represented similarly while trucks and motorcycles will be represented differently. An example representation by the SP is shown in Figure 3.

The task of the TM will then be to learn the common patterns that the cars exhibit, their speed, shape, and positioning will be taken into account. Finally, the learning will be set so that new patterns are learned quickly, but forgotten slowly. This will allow the model to quickly learn the norm, even if there is little activity, while still reacting to anomalies. This requires that the point of view is stationary, in our example this means that the camera is stationary.

It is possible to split different segmentation classes into separate SDRs. This will give the SP and the TM the ability to learn different behaviors for each of the classes. For instance, if there are “person” and “car” classes, then, the TM will learn that it is normal for objects belonging to “person” to be on the sidewalk, while objects belonging to “car” will be marked as anomalous when on the sidewalk.

Ideally, the architecture will have a calibration period spanning several days or weeks, during which the architecture is not performing any anomaly detection, but is just learning the patterns. In general, the duration of the calibration period is set so that the most common events have had time to be observed.

## 4. Improvements

Daylidyonok et al. [25] tested only the base HTM algorithm and showed that the algorithm cannot handle subtle anomalies, and therefore multiple improvements needed to be introduced to increase effectiveness.

### 4.1. Invariance

One issue that becomes evident is the lack of invariance, due to the TM learning the global patterns. Using the example in Figure 2, it learns that it is normal for cars to drive along the road but only in the context of there being cars parked in the parking lot. It is instead desired that the TM learns that it is normal for cars to drive along the road, regardless of whether there are cars in the parking lot. We propose a solution based on dividing the encoder output into a grid and have a separate SP and TM for each cell in the grid. The anomaly scores of all the cells are then aggregated into a single anomaly score using an aggregation function.

### 4.2. Aggregation Function

Selecting the correct aggregation function is important because it affects the final anomaly output. For instance, it might be tempting to use the mean of all the anomaly scores as the aggregation function:$$\begin{array}{cc} X:\{x\in R:x\geq 0\}\\ Anomaly\_Score & =\frac{\sum_{x\in X}x}{\left|X\right|} \end{array}$$
where *X* denotes the set of anomaly scores *x* from each individual grid. However, this leads to problems with normalization, meaning that an overall anomaly score of 1 is hard to achieve due to many cells having a zero anomaly score. In fact, it becomes unclear what a high anomaly score is anymore. Using the mean also means that anomalies that take up a lot of space will be weighted higher than anomalies that take up little space, which might not be desirable. To solve the aforementioned problem and given that the data contain little noise, a potential aggregation function could be the non-zero mean:$$\begin{array}{cc} X & :\{x\in R:x>0\}\\ Anomaly\_Score & =\left\{\begin{array}{cc} \frac{\sum_{x\in X}x}{\left|X\right|} & \mathrm{if}\;|X|>0\\ 0 & \mathrm{otherwise} \end{array}\right. \end{array}$$

This means that only the cells with a strictly positive anomaly score will be contributing to the overall anomaly score which helps solve the aforementioned problems. Meanwhile, the non-zero mean will perform poorly when the architecture is exposed to noisy data which could lead to there always being cells with a high anomaly score. To clarify, noise in this context refers to unknown events occurring that are not necessarily anomalies, such as flickering due to objects falling above/below the instance classification threshold, or randomness in the input data such as sperm cells that vibrate seemingly randomly.

Figure 4 illustrates the effect of an aggregation function for noisy data, where the non-zero mean is rendered useless due to the noise. On the other hand, Figure 5 shows how the non-zero mean gives a clearer anomaly score when the data is clean compared to the noise data which has a more chaotic line.

### 4.3. Explainability

Having the encoder output divided into a grid has the added benefit of introducing explainability into the model. By using Grid HTM, it is now possible to determine where in the input an anomaly has occurred by simply observing which cell has a high anomaly score. It is also possible to estimate the number of predictions for each cell [29] which can be used as a measure of certainty, where fewer predictions means higher certainty. Making it possible to measure certainty per cell creates a new source of information that can be used for explainability or robustness purposes.

### 4.4. Flexibility and Performance

Furthermore, it is also possible to configure the SP and the TM in each cell independently, giving the architecture increased flexibility, and to use a non-uniform grid, meaning that some cells can have different sizes. Furthermore, dividing the frame into smaller cells makes it possible to run each cell in parallel for increased performance.

### 4.5. Reviewing Encoder Rules

A potential challenge with the grid approach is that the rules [12] for creating a good encoder, may not be respected and therefore should be reviewed:Semantically similar data should result in SDRs with overlapping active bits. In this example, a car at one position will produce an SDR with a high amount of overlapping bits as another car at a similar position in the input image.The same input should always produce the same SDR. The segmentation model produces a deterministic output given the same input.The output must have the same dimensionality (total number of bits) for all inputs. The segmentation model output has a fixed dimensionality.The output should have similar sparsity (similar number of one-bits) for all inputs and have enough one-bits to handle noise and subsampling. The segmentation model does not respect this. An example is that there can be no cars (zero active bits), one car (*n* active bits), or two cars (2n active bits), and that this will fluctuate over time.

The solution for the last rule is two-fold, and consists of imposing a soft upper bound and a hard lower bound for the number of active pixels within a cell. The purpose is to lower the variation of number of active pixels, while also containing enough semantic information for the HTM to work:Pick a cell size so that the distribution of number of active pixels (white pixels, representing active bits) is as tight as possible, while containing enough semantic information and also being small enough so that the desired invariance is achieved. The cell size acts as a soft upper bound for the possible number of active pixels.Create a pattern representing emptiness, where the number of active bits is similar to what can be expected on average when there are cars inside a cell. This acts as a hard lower bound for the number of active pixels.

There could be situations where a few pixels are active within a cell, which could happen when a car has just entered a cell, but this is acceptable as long as it does not affect the distribution too much. If it does affect the distribution, which can be the case with noisy data, then an improvement would be to add a minimum sparsity requirement before a cell is considered not empty, e.g., less than five active pixels means that the cell is empty. In the following example, the number of active pixels within a cell centered in the video was used to build the distributions as seen in Figure 6.

With a carefully selected empty pattern sparsity, the standard deviation of active pixels was lowered from 3.78 to 1.41. It is possible to automate this process by developing an algorithm which finds the optimal cell size and empty pattern sparsity which causes the least variation of the number of active pixels per cell. This algorithm would run as a part of the calibration process.

In addition to the aggregated anomaly score, the visual representation of these changes is also a crucial output. This can be observed in Figure 7, where cells are color-coded with red indicating a higher anomaly score and green indicating a lower anomaly score.

### 4.6. Stabilizing Anomaly Output

An additional limitation of the grid-based approach is its inability to anticipate the presence of a car entering a cell. The TM within a cell is unable to detect external factors, resulting in a high anomaly output when a car first enters a cell. This is illustrated in Figure 8, where it can be observed that this effect causes the anomaly output to needlessly fluctuate. The band-aid solution is to ignore the anomaly score for the frame during which the cell goes from being empty to being not empty, as illustrated in Figure 9.

A more proper solution could be to allow the TM to grow synapses to the TMs in the neighboring cells; however, this is not documented in any research papers and might also hinder invariance. A pyramid-based architecture could also be explored for solving this issue.

### 4.7. Multistep Temporal Patterns

Since the TM can only grow segments to cells that were active in the previous timestep, it will struggle to learn temporal patterns across multiple timesteps [30]. This is especially evident in high-frame-rate videos, where an object in motion has a similar representation at timestep *t* and t+1, as an object standing still.

This could cause situations where an object that is supposed to be moving, suddenly halts, yet the TM will not mark it as an anomaly due to it being stuck in a contextual loop. A contextual loop occurs when a prediction made at time *t* is fulfilled by the state at time t+1, and then the prediction made at time t+1 is similar to the state at time *t*, which in this example happens if the object stops moving. This leads the TM to repeat the same state it was in at time *t*, creating a loop.

To address this, one solution is to incorporate the past *n* SP outputs as input to the TM by maintaining a buffer of past SP outputs and updating it as new SP outputs are received. This is illustrated in Figure 10.

This follows the core idea behind encoding semantic time in addition to the data, which makes time, such as time of day and day of the week, act as a contextual anchor. However, in this example there are no cyclical time elements that are suitable to be used as contextual anchors, so as a replacement, the past observations are encoded instead.

Concatenating past observations together will force the TM input, for when an object is in motion and when an object is still, to be unique. High-frame-rate videos can benefit the most from this, and the effect will be more pronounced for higher values of *n*.

An advantage of incorporating temporal patterns is that it may increase the robustness of the TM to temporal noise. For example, if an object is not captured by the deep learning segmentation model encoder for a single frame, and thus disappears from the scene, this noise will have less impact on the TM’s input as it will only constitute a small fraction $\frac{1}{n}$ of the input, as the TM is now exposed to multiple frames at once.

### 4.8. Use Cases

One of the most straightforward applications of GridHTM is in semi-active surveillance, where personnel are only required to examine the segments that contain anomalies, resulting in a significant improvement in efficiency. For example, it could enable monitoring of an entire city with a minimal number of personnel. This is achieved by directing attention to only the anomalous segments identified by GridHTM, which greatly reduces the manpower required for active monitoring of the entire city.

## 5. Experimental Details and Results

As stated earlier, one of the use cases of GridHTM is anomaly detection in surveillance, and using a video from the VIRAT [28] video dataset with long duration and a stationary camera, we demonstrate our system. The video contains technical anomalies in the form of sudden frame skips, as well as a synthetic anomaly of a frame repeat lasting several seconds, which is included to test the ability of GridHTM to detect unusual temporal patterns and understand the expected movement of objects over time. The working code for GridHTM and the parameters for the experiments conducted in for this paper can be found on GitHubhttps://github.com/vladim0105/GridHTM (accessed on 10 February 2023).

In this experiment, a segmentation model that can extract classes into their respective SDRs is employed. This means that there could be an SDR for cars and an SDR for pedestrians, that are then concatenated before being fed into the system.

The segmentation model used is PointRend [31] with a ResNet101 [32] backbone, pretrained on ImageNet [33], and implemented using PixelLib [34]. For the sake of simplicity, this experiment will focus only on the segmentation of cars. While on the topic of segmentation, it is important to mention that the segmentation model is not perfect and that there are cases where objects are misclassified as well as cases where cars repeatedly go above and below the confidence threshold. It should also be mentioned that the system trains as it observes, thus no pre-training of the system was performed before applying it to the video, unlike the typical approach used with deep learning systems.

Normally, one would compare the performance of the system with state-of-the-art deep learning methods, but at the time of writing there are no deep learning systems that operate on the same premises and can provide a fair comparison. Typically, unsupervised deep learning approaches are pre-trained on a specific domain and therefore perform poorly when dealing with other domains or drifting changes in a domain, whereas GridHTM can be put in potentially any domain and will learn its behaviors online.

### 5.1. Results

We can see in Figure 11 that GridHTM is detecting when segments begin and end. However, it is not possible to use a threshold value to isolate them, and they also have vastly different anomaly scores compared to each other. This is due to the way the aggregation function works, which means that the anomaly output is dependent on the physical size of the anomaly. It should also be noted that a moving average (n=200) was applied to smooth out the anomaly score output, otherwise the graph would be too noisy.

With the aggregation functions presented in this paper in mind, it is safe to conclude that looking at the anomaly score output is meaningless for complex data such as a surveillance video. This however does not mean that GridHTM is completely useless, and this can be observed by looking at the visual output of GridHTM. The visual output during which the first segment anomaly occurs can be observed in Figure 12. Here, it is observed that GridHTM correctly marks the sudden change of cars when the current segment ends and a new segment begins.

#### 5.1.1. Road

In the original video, there is a road on which cars regularly drive. By observing the visual output, it becomes evident that after some time GridHTM has mostly learned that behavior and does not report those moving cars as anomalies. This is shown in Figure 13.

#### 5.1.2. Frame Repeat

To prove that GridHTM has learned that cars on the road should be moving, it is possible to inspect the visual output during the period when the video is repeating the same frame and observe if the system identifies the cars that are stationary on the road as anomalies.

It can be observed in Figure 14 that the cars along the main road are not marked as anomalies, but this could be attributed to the fact that there is a crossing there and that cars periodically have to stop at that point to let pedestrians cross.

On the other hand, the anomaly marked with a blue circle shows a car on the road in a parking lot that is identified as an anomaly, and the anomaly’s severity increases as the frame is repeated. The reason for this is that, unlike the cars on the main road, it is unusual to see a car stationary in that position, thus causing the anomaly. To confirm that the anomaly was specifically caused by the repeated frame and not simply a repetition of the anomaly over time, it should be compared to the anomaly output in the absence of the repeated frame.

As shown in Figure 15, it can be observed that the anomaly output is much lower when there is no repeating frame. This confirms that the anomaly was caused by the repeated frame, and that GridHTM was able to learn the expected temporal patterns of moving objects.

It is also noteworthy to examine how GridHTM handles the repeated frames without incorporating multistep temporal patterns, the result of which is shown in Figure 16.

Unfortunately, simply disabling multistep temporal patterns without adjusting the other TM parameters causes the same car to be marked as an anomaly both before and during the repeated frame. In fact, as previously mentioned, disabling multistep temporal patterns causes GridHTM to become less noise tolerant, thus causing many more anomalies to be wrongly detected. This can be seen in Figure 16, where a higher number of severe anomalies are observed compared to previous examples. This also illustrates how sensitive HTM can be with regard to parameters.

## 6. Conclusions

In this work, we presented a novel method called Grid-based Hierarchical Temporal Memory (GridHTM) for anomaly detection in videos. The experimental results demonstrate that the proposed GridHTM has potential for unsupervised anomaly detection in complex videos such as surveillance footage. Notably, the method proposed herein is an extension of traditional HTM which is usually only able to handle single data input streams. This insight could also be used to extend the standard HTM even further in future work.

One of the most important future works would be to create a dataset with videos that are several days long and contain anomalies such as car accidents, jaywalking, and other similar anomalous behaviors. For GridHTM, more time can be spent exploring other aggregation functions so that the aggregated anomaly score can be used more efficiently. One could employ deep learning for this purpose or perhaps use another layer of HTM, the possibilities are endless. Another area of improvement would be to develop an algorithm that can automatically set the parameters for each cell during the calibration phase. Additionally, it would be beneficial to increase explainability and robustness by implementing a measure of certainty for each cell. Moreover, experiments should be conducted to evaluate the use of depth or 3D vision for anomaly detection in surveillance, as the depth information could be valuable. This could be achieved by using voxels, which can be employed similarly to 2D segmentation, where an extra SDR could be created for each layer of depth in the voxelized 3D image. Furthermore, to address the problem of unstable anomaly output, it might be worthwhile to investigate the possibility of having the TM in each cell form synapses with neighboring cells.

## Figures and Tables

**Figure 1 sensors-23-02087-f001:**
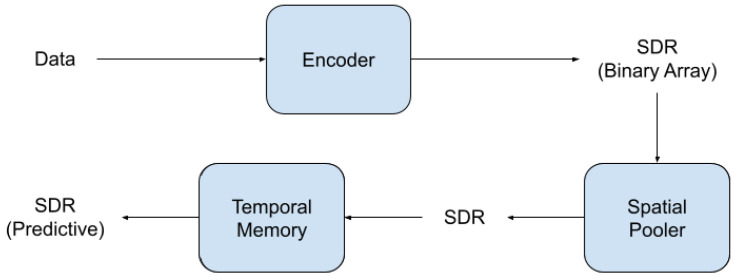
A typical HTM pipeline. A common next-step could be to use a classifier to convert the predictive SDR into a classification, or to compare the prediction with the actual outcome.

**Figure 2 sensors-23-02087-f002:**
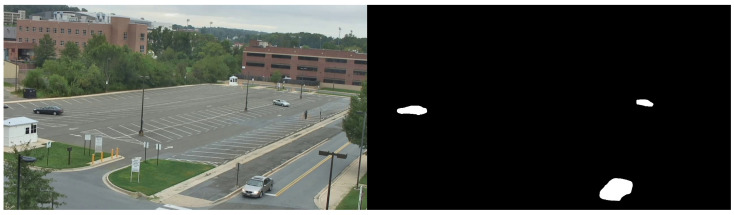
Segmentation result of cars, which is suited for use as an SDR. Original frame taken from VIRAT [28].

**Figure 3 sensors-23-02087-f003:**
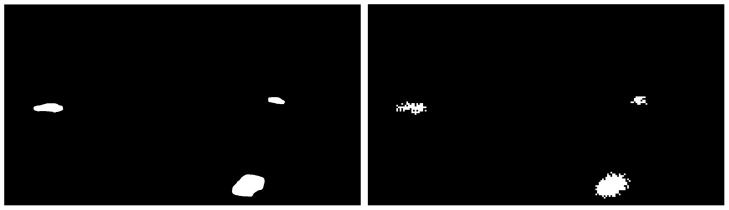
The SDR (**left**) and its corresponding SP representation (**right**) for the original image in Figure 2. Both are SDRs, but the one produced by the SP is of a lower dimension and tries to capture unique features that it observes are reoccurring.

**Figure 4 sensors-23-02087-f004:**
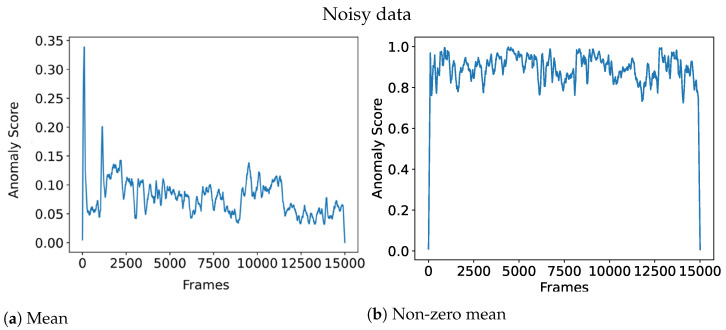
Aggregation function performance on noisy data.

**Figure 5 sensors-23-02087-f005:**
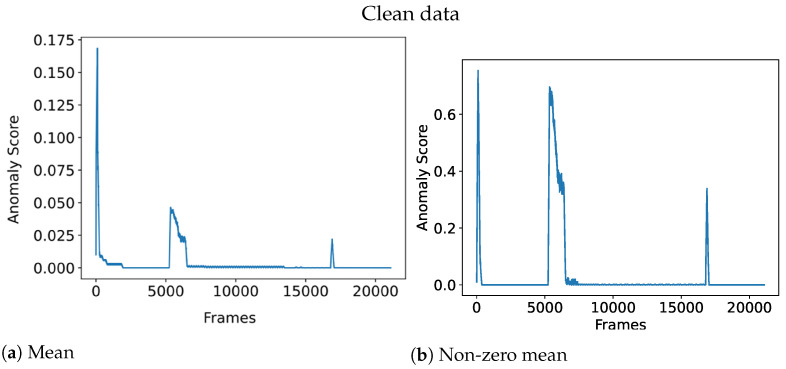
Aggregation function performance on clean data.

**Figure 6 sensors-23-02087-f006:**
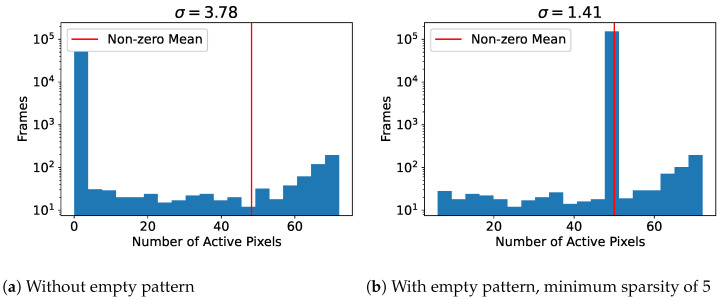
Distribution of number of active pixels within a cell of size 12×12.

**Figure 7 sensors-23-02087-f007:**
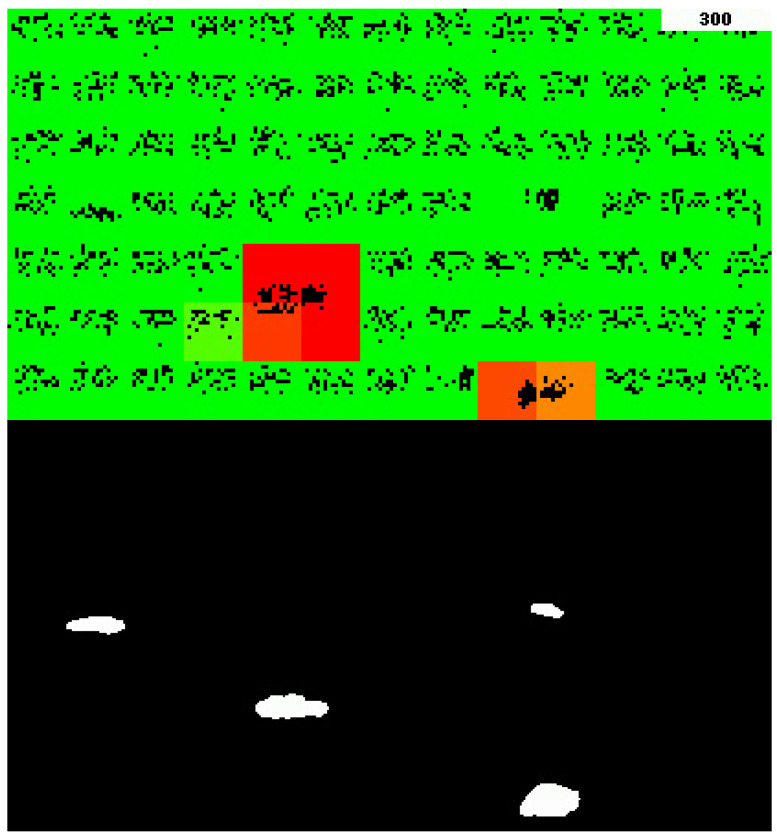
Example Grid HTM output and the corresponding input. The color represents the anomaly score for each of the cells, where red represents high anomaly score and green means zero anomaly score. Two of the cars are marked as anomalous because they are moving, which is something Grid HTM has not seen before during its 300 frame-long lifetime (top right).

**Figure 8 sensors-23-02087-f008:**
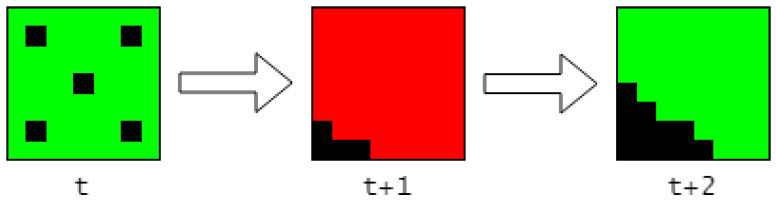
High anomaly score when an empty cell (represented with an empty pattern with a sparsity value of 5) changes to being not empty, as something enters the cell.

**Figure 9 sensors-23-02087-f009:**
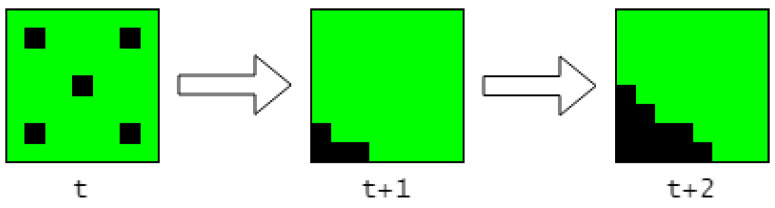
The anomaly score is ignored (set to 0) for the frame in which the cell changes state from empty to not empty.

**Figure 10 sensors-23-02087-f010:**
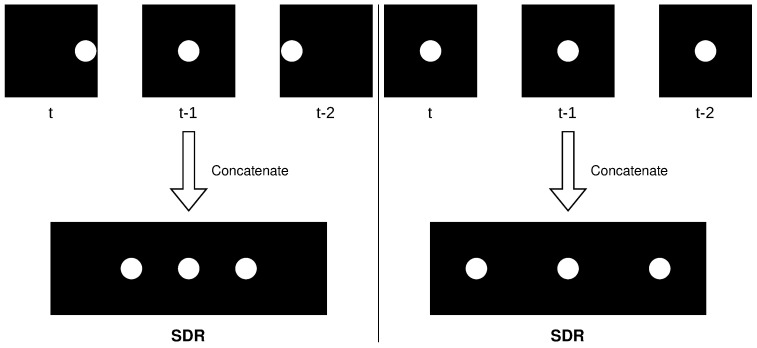
Example of concatenation with n=3 when an object is moving from left to right (**left**), compared to when an object is not in motion (**righ**t). It can be observed that the SDRs are vastly different.

**Figure 11 sensors-23-02087-f011:**
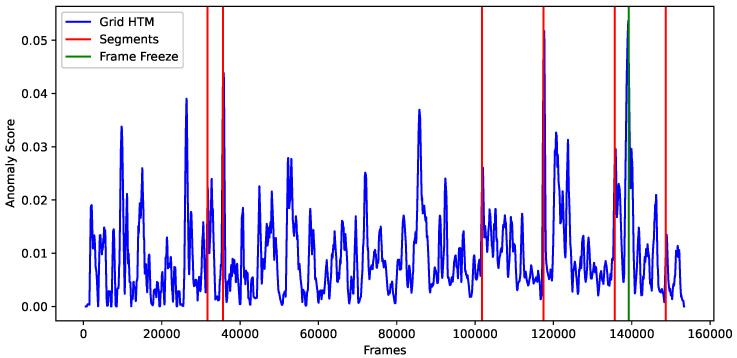
Anomaly score output from GridHTM.

**Figure 12 sensors-23-02087-f012:**
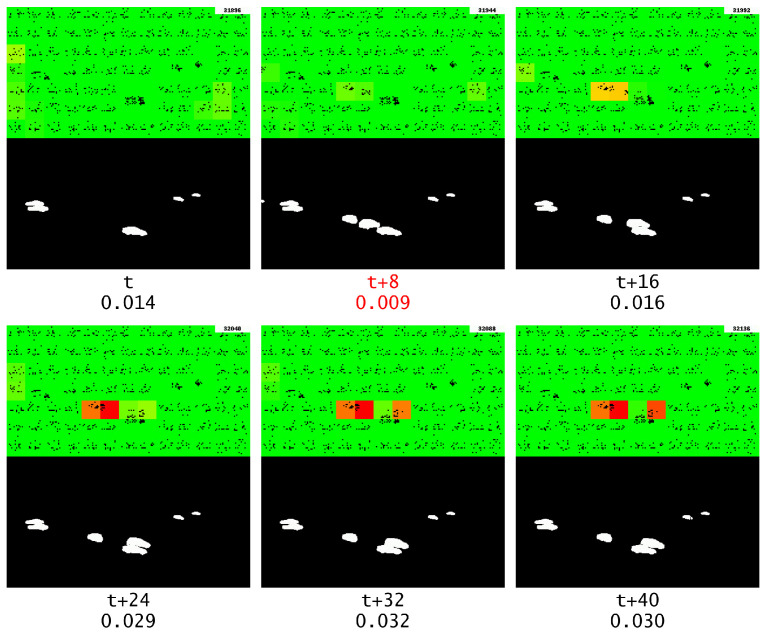
The first segment anomaly, which is marked with red text, and the corresponding changes detected by GridHTM (from red to yellow, strong to light weak change and green no change). The numbers beneath each frame represent the relative frame number and the current anomaly score, respectively.

**Figure 13 sensors-23-02087-f013:**
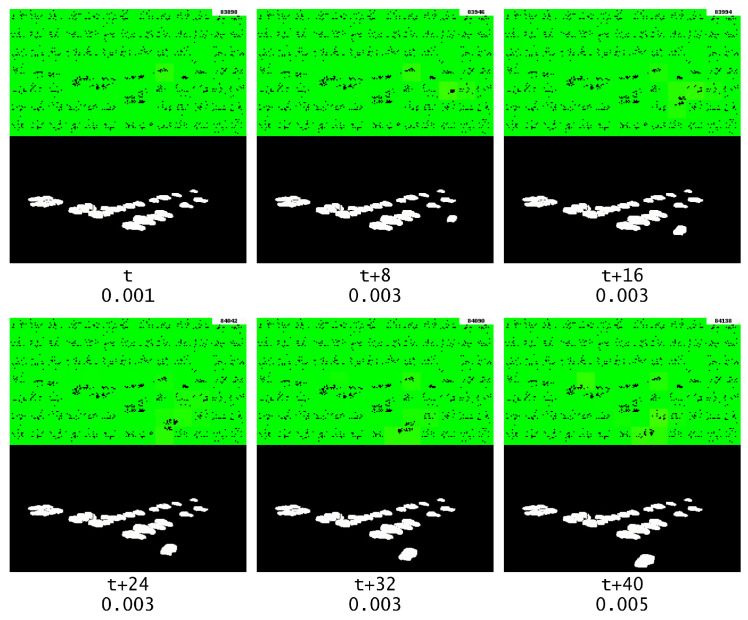
Visual output of a car driving along a road.

**Figure 14 sensors-23-02087-f014:**
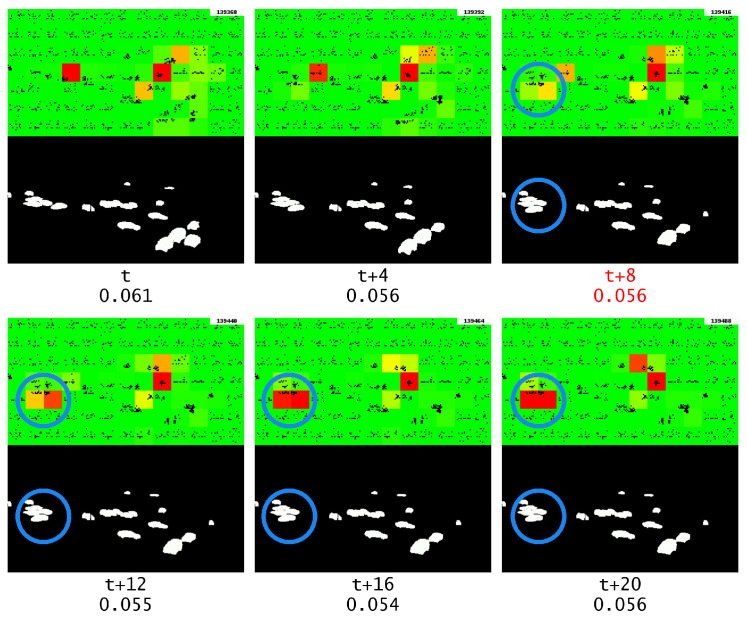
Anomaly output during the repeating frame, the start of the frame repeat is marked with red text. The blue circle highlights the object of interest.

**Figure 15 sensors-23-02087-f015:**
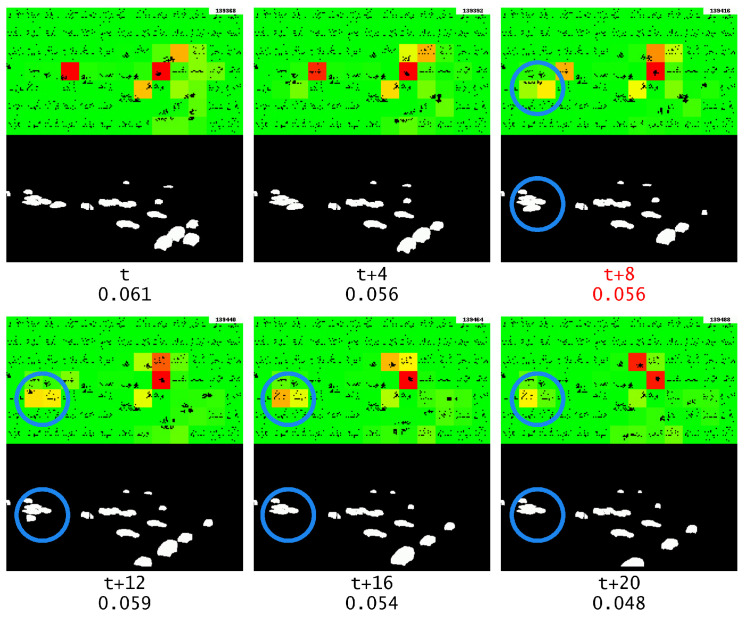
Anomaly output when there is no frame repeating, where it should have repeated is marked with red text. The blue circle highlights the object of interest.

**Figure 16 sensors-23-02087-f016:**
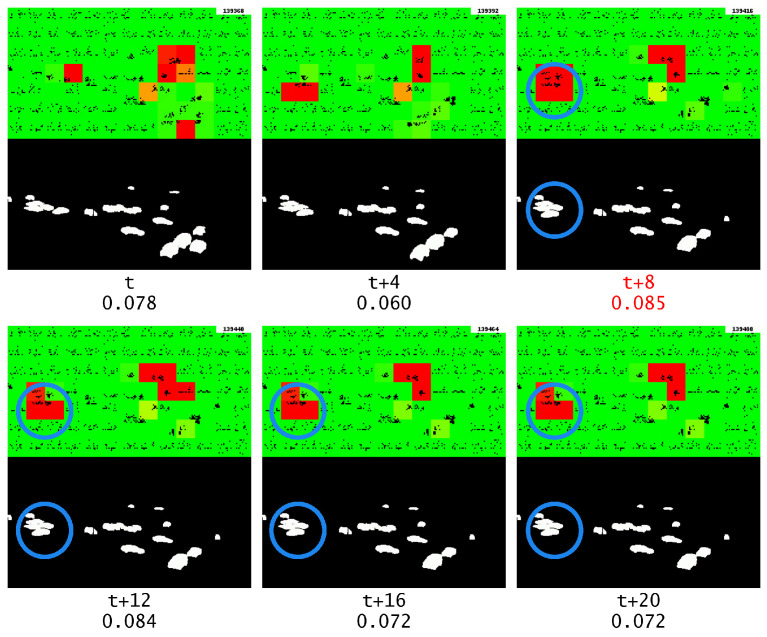
Anomaly output during the repeating frame, the start of the frame repeat is marked with red text. The blue circle highlights the object of interest. This time without multistep temporal patterns.

## Data Availability

Not applicable.

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
