# Peer review of "GridHTM: Grid-Based Hierarchical Temporal Memory for Anomaly Detection in Videos"

_sensors, 2023, doi:10.3390/s23042087_

Round 1
Reviewer 1 Report
The paper is interesting and the results are compelling, but a clear apples to apples comparison was not presented. When presenting new ML methods there needs to be some way of measuring the performance of the new method against existing methods. If there are no other unsupervised methods to compare against then this point needs to be made more clear. Also the presentation needs to be cleaned up, there are missing figures and figures with cutoff axis labels. The missing figures must be included in order to make a complete evaluation of the manuscript.
Author Response
Dear reviewer,
thanks a lot for your input. We have made more clear why at this stage a comparison with other methods does not make sense. In addition we cleaned up the presentation and improved the overall readability and quality of the article.

Reviewer 2 Report
The paper requires significant improvements, from English language usage to method presentation. The abstract misses important information like summaries of findings, testing data description, and results. The introduction is incomplete and misses the proposed work contributions, testing data overview, and achieved results. Moreover, the motivations behind the research are weak; and currently, deep learning approaches can handle noise and concept drift with ad-hoc strategies. However, outliers (i.e., anomalies) generally do not imply a change in the underlying data distribution or regime. A Related Works section is completely missing, even if the literature on HTM applied to Anomaly Detection and the latter applied to surveillance videos exist. This choice of not including an overview of existing works is unexplainable; it should be used to highlight and support the proposed method contributions. However, it is hard to follow the method and the proposed improvements; the background materials required to understand the novelty are missing, and the presented content is hard-to-read and follow. Figure labels are unclear. What does [t]0.5 stand for?. Moreover, sometimes the figure position is indicated as “left” or “right” when they are top or bottom. In general, there are several writing style problems. Concerning experiments, they should be extended. For example, authors can execute public SoTA deep learning models for anomaly detection on the same datasets, comparing the results.
Author Response
Dear reviewer,
thank you very much for your input. We have gone trough the article and improved on the points you mentioned. We also added more discussion and explanations about why at this stage of the research a comparison with deep learning methods is not meaningful.

Reviewer 3 Report
The paper explores the capabilities of the Hierarchical Temporal Memory (HTM) algorithm to perform anomaly detection in videos, as it has favorable properties such as noise tolerance and online learning, which combats concept drift. The experimental results demonstrate the effectiveness of the approach. The topic is valuable, but some points should be improved.
1. In the abstract section, the principal objectives and scope of the investigation need to be better defined. Why do you select GridHTM? In addition, I suggest summarizing the results and stating the principal conclusion.
2. In lines 21-22, you state, "The most important component in video anomaly detection systems is the intelligence behind it." I do not quite agree with it because intelligence is the way to detect video anomalies, not the important component.
3. The Introduction is too weak. It should state the GridHTM in more detail and propose the reasons for the choice of this method.
4. Concerning the background, you should summarize the advantages and disadvantages of related video anomaly detection methods and compare the proposed approach with the existing methods.
5. In describing GridHTM, you should direct readers to sufficient details so they can repeat the experiments. What's the key innovation of your method compared to related works?
6. On page 6, there exists Figures 4 and 5; perhaps it's an error.
7. In lines 141-147, you propose the Explainability. However, figure 5 cannot evaluate the Explainability. More justification is needed for this claim.
8. The experimental results need to be further discussed because it is the contribution of your works.
9. In the Discussion section, it's better to summarize your evidence for each conclusion and state your conclusions as clearly as possible.
10. There are many expression errors, indicating that the paper needs to be carefully checked for these errors.
Author Response
Dear reviewer,
thanks you very much for the detailed comments. We have addressed all of them for the revised version.
Some of the main changes are: - Added some more details to HTM background, including figures to help conceptualize what is written. Also added a disclaimer, that says that we chose to keep a high-level overview of HTM for the sake of brevity - Added more citations to both address the critiques and to support more statements - Various rewrites and information-gap filling

Round 2
Reviewer 1 Report
No additional comments at this time.
Author Response
Dear reviewers,
Thank you very much for your valuable comments and time spent reviewing our manuscript. Here, we have done our best to address your comments or give clear explanations for your comments. We also went through the whole paper again to improve English language and formatting. Point-by-point responses are given below.
Reviewer 01 comments:
The first reviewer was happy with our revision and had “no additional comments at this time”.
Reviewer 02 comments:
These issues reported in the previous review report are still present:
- the abstract misses important information like summaries of findings, testing data description, and results;
We made changes to the abstract that better reflects the content and goal of the work.
- the introduction needs to be completed and include the proposed work contributions, testing data overview, and achieved results. Moreover, the motivations behind the research could be stronger; currently, deep learning approaches can handle noise and concept drift with ad-hoc strategies. However, outliers (i.e., anomalies) generally do not imply a change in the underlying data distribution or regime;
We agree on the statement for the outliers and think this might be a misunderstanding and not clear enough, thus we tried to make it more clear in the background section of the work. In addition we added some discussion on the raised points in the introduction and related work. We also added specific contributions of our work to the introduction.
- Background section should be renamed in Related Works and, other than cited, authors should briefly discuss each reported work;
We renamed the section and added discussion points where we thought it was relevant.
- The authors claim that their method overcomes some deep learning method challenges. However, they must find a way or dataset suitable to quantify and prove it. Moreover, authors can also run SoTA deep learning methods on the same dataset quantifying the performances (e.g., AUC measure) and comparing them independently from premises. The latter shows how good their model is in discriminating between classes (i.e., normal and anomaly) with respect to recent trends.
We do not see a direct comparison as important for this paper. This work is mainly focused on exploring the feasibility of using HTM for visual content and making it more accessible for the community. Comparing it with different deep learning methods would not add important information to the paper and is rather something for future work. We added argumentations for this into the introduction and related work.
Reviewer 03 comments:
The paper has improved significantly, and I recommend minor revisions/proofreading.
We have performed minor revisions and performed a proofreading correcting several minor language errors.
Reviewer 2 Report
These issues reported in the previous review report are still present:
- the abstract misses important information like summaries of findings, testing data description, and results;
- the introduction needs to be completed and include the proposed work contributions, testing data overview, and achieved results. Moreover, the motivations behind the research could be stronger; currently, deep learning approaches can handle noise and concept drift with ad-hoc strategies. However, outliers (i.e., anomalies) generally do not imply a change in the underlying data distribution or regime;
- Background section should be renamed in Related Works and, other than cited, authors should briefly discuss each reported work;
- The authors claim that their method overcomes some deep learning method challenges. However, they must find a way or dataset suitable to quantify and prove it. Moreover, authors can also run SoTA deep learning methods on the same dataset quantifying the performances (e.g., AUC measure) and comparing them independently from premises. The latter shows how good their model is in discriminating between classes (i.e., normal and anomaly) with respect to recent trends.
Author Response

(The authors gave the same response as above.)

Reviewer 3 Report
The paper has improved significantly, and I recommend minor revisions/proofreading.Author Response
Dear reviewers,
Thank you very much for your valuable comments and time spent reviewing our manuscript. Here, we have done our best to address your comments or give clear explanations for your comments. We also went through the whole paper again to improve English language and formatting. Point-by-point responses are given below.
Reviewer 01 comments:
The first reviewer was happy with our revision and had “no additional comments at this time”.
Reviewer 02 comments:
These issues reported in the previous review report are still present:
- the abstract misses important information like summaries of findings, testing data description, and results;
We made changes to the abstract that better reflects the content and goal of the work.
- the introduction needs to be completed and include the proposed work contributions, testing data overview, and achieved results. Moreover, the motivations behind the research could be stronger; currently, deep learning approaches can handle noise and concept drift with ad-hoc strategies. However, outliers (i.e., anomalies) generally do not imply a change in the underlying data distribution or regime;
We agree on the statement for the outliers and think this might be a misunderstanding and not clear enough, thus we tried to make it more clear in the background section of the work. In addition we added some discussion on the raised points in the introduction and related work. We also added specific contributions of our work to the introduction.
- Background section should be renamed in Related Works and, other than cited, authors should briefly discuss each reported work;
We renamed the section and added discussion points where we thought it was relevant.
- The authors claim that their method overcomes some deep learning method challenges. However, they must find a way or dataset suitable to quantify and prove it. Moreover, authors can also run SoTA deep learning methods on the same dataset quantifying the performances (e.g., AUC measure) and comparing them independently from premises. The latter shows how good their model is in discriminating between classes (i.e., normal and anomaly) with respect to recent trends.
We do not see a direct comparison as important for this paper. This work is mainly focused on exploring the feasibility of using HTM for visual content and making it more accessible for the community. Comparing it with different deep learning methods would not add important information to the paper and is rather something for future work. We added argumentations for this into the introduction and related work.
Reviewer 03 comments:
The paper has improved significantly, and I recommend minor revisions/proofreading.
We have performed minor revisions and performed a proofreading correcting several minor language errors.
